# Impact of COVID-19 in Children with Chronic Lung Diseases

**DOI:** 10.3390/ijerph191811483

**Published:** 2022-09-13

**Authors:** Valentina Agnese Ferraro, Stefania Zanconato, Silvia Carraro

**Affiliations:** Unit of Pediatric Allergy and Respiratory Medicine, Women’s and Children’s Health Department, University of Padova, 35128 Padova, Italy

**Keywords:** COVID-19, SARS-CoV-2, chronic lung disease, children

## Abstract

Background: since December 2019, the world has become victim of the coronavirus disease 2019 (COVID-19) pandemic caused by severe acute respiratory syndrome coronavirus-2 (SARS-CoV-2). The aim of our narrative review is to analyze the impact of COVID-19 in children suffering from chronic lung disease (CLD). Methods: we searched the MEDLINE/Pubmed database using the terms “SARS-CoV-2” or “COVID-19” or “Coronavirus Diseases 2019”; AND “chronic lung diseases” or “chronic respiratory diseases” or “asthma” or “cystic fibrosis” or “primary ciliary dyskinesia” or “bronchopulmonary dysplasia”; and limiting the search to the age range 0–18 years. Results and Conclusions: although COVID-19 rarely presents with a severe course in children, CLD may represent a risk factor; especially when already severe or poorly controlled before SARS-CoV-2 infection. On the other hand, typical features of children with CLD (e.g., the accurate adoption of prevention measures, and, in asthmatic patients, the regular use of inhaled corticosteroids and T2 inflammation) might have a role in preventing SARS-CoV-2 infection. Moreover, from a psychological standpoint, the restrictions associated with the pandemic had a profound impact on children and adolescents with CLD.

## 1. Introduction

Since December 2019, the whole world has become victim of the coronavirus disease 2019 (COVID-19) pandemic caused by severe acute respiratory syndrome coronavirus-2 (SARS-CoV-2). COVID-19 is mainly considered an “adult pandemic”, the lungs being the main organ involved. Compared with adults, children were less heavily affected by this illness; which has a lower incidence in the pediatric population, milder symptoms, lower mortality (about 0–0.2%), and a better prognosis [1,2]. Common symptoms in COVID-19 infection in children are: low grade fever, cough, runny nose and upper respiratory tract involvement, abdominal pain, and diarrhea [3,4]. After the acute phase, children could develop a serious condition with systemic inflammation called multisystem inflammatory syndrome in children (MISC); the main features of which are fever, multiorgan dysfunction, and elevated acute phase reactants [5].

In addition to the direct effects of COVID-19 in children, we must keep in mind the indirect effects on children’s physical and psychological functioning due to quarantine policies, temporary closure of schools and leisure activities, and distance teaching [6].

Children with chronic lung diseases (CLD) represent a vulnerable population for several reasons: they could be at risk of a severe form of COVID-19 because of their underlying chronic condition; the psychological impact of the pandemic could be stronger; and the usual management and follow-up of their disease could be disrupted.

The aim of our narrative review is to analyze the impact of COVID-19 in children with chronic lung disease (CLD).

## 2. Materials and Methods

We identified relevant published studies by searching the MEDLINE/Pubmed database using the following terms: “SARS-CoV-2” or “COVID-19” or “Coronavirus Diseases 2019”; AND “chronic lung diseases” or “chronic respiratory diseases” or “asthma” or “cystic fibrosis” or “primary ciliary dyskinesia” or “bronchopulmonary dysplasia”. The filters applied were: language (English) and age of the study subjects (0–18 years).

## 3. Children with CLD and Risk for Severe COVID-19

According to a multicenter cohort study by Götzinger et al. [7] reported in September 2020, in 21 European countries, there were 582 cases of SARS-CoV-2 infection in children; of whom 8% developed a severe disease, with the following risk factors for ICU admission: an age of less than 1 month; the presence of at least one pre-existing medical condition (including CLD); male sex; and the presence of lower respiratory tract infection signs or symptoms at presentation. In keeping with this, in October 2020, Bellino et al. [8] showed that only 4.3% of 3836 Italian children had a severe COVID-19 course; with a higher risk reported in patients with preexisting underlying medical conditions, including respiratory diseases (odds ratio 2.80; 95% CI 1.74–4.48). Furthermore, a larger study published by Kompaniyets et al. [9] on more than 43,000 children in the United States from March 2020 to January 2021, showed that 9.9% of affected children were hospitalized and 29.6% of them were admitted to the ICU. Even if the most frequent diagnosed condition in this cohort was asthma, it was not found to be associated with a higher risk of COVID-19 illness among hospitalized children; except among those aged 12 to 18 years [9].

Focusing on children with pre-existing respiratory conditions, such as asthma, cystic fibrosis (CF), or BPD, a survey conducted by the Pediatric Assembly of the European Respiratory Society (ERS) [10], published in December 2020, analyzed the burden of hospitalizations. It found that only a low number of children were admitted to hospital, and only a minority of children with BPD and other CLD required ventilatory support; this suggests that these children are not at increased risk for severe COVID-19. Moreover, Papadopoulous et al. [11] showed in the PeARL multinational cohort (1054 asthmatic children and 505 non-asthmatic) that asthmatic children experienced, during the first wave of the COVID-19 pandemic, fewer respiratory infections, emergency visits, hospital admissions, and asthma attacks in comparison with the previous year. Such improved asthma control, probably due to reduced exposure to asthma triggers and increased treatment adherence, led by the fear of COVID-19, points out that asthma was not a risk factor for severe COVID-19. In keeping with this, a retrospective cohort study [12] in North Carolina found no evidence that asthma predisposes children to SARS-CoV-2 infection or severe illness from COVID-19. However, the level of asthma control is a key factor with respect to the risk of severe COVID-19. In fact, a recent cohort study in Scotland showed that school-aged children with poorly controlled asthma (i.e., previous asthma hospital admission, or two or more courses of oral corticosteroids in the previous 24 months) have a markedly increased risk of COVID-19 hospital admission [13].

An international longitudinal study [14] on 640 patients with primary ciliary dyskinesia (PCD) found that only 3.4% of the pediatric population (8 children out of the 234 patients, aged 0–19 years) had a SARS-CoV-2 infection; with only one child hospitalized and no ICU admission. Moreover, the CF Registry Global Harmonization Group, collecting data from 13 countries between February and August 2020 [15], showed that 22.9% of the 105 children infected were admitted to hospital. The hospitalized children had a severe pre-existing lung disease; however, only six required supplementary oxygen and two required non-invasive ventilation.

In summary, given that COVID-19 rarely presents with a severe course in children, CLD may represent a risk factor; especially when already clinically severe or poorly controlled before SARS-CoV-2 infection (Figure 1). It is evident from this that children with CLD should be considered a priority for vaccination against SARS-CoV-2 [13].

## 4. How CLD Could Affect the Risk of SARS-CoV-2 Infection

The risk of SARS-CoV-2 infection and of the course of COVID-19 may be affected by three features typical of patients with CLD: the attitude toward respiratory infection prevention; in the case of asthmatic patients, the regular use of inhaled steroids (ICS); and the underlying allergic inflammation (Figure 1).

Patients and parents with CLD are more likely to practice careful protective behaviors in order to avoid respiratory infections; these behaviors include hand hygiene, mask wearing, and social distancing [16]. This conduct has been reinforced during the first wave of COVID-19 by North-American and European recommendations, which advised to prevent pulmonary exacerbations [16,17,18].

Secondly, in patients with CLD, the risk and course of SARS-CoV-2 infection could be affected by maintenance therapy; and, in particular, by ICS therapy that is extensively used in the treatment of asthma. In vitro studies on the impact of ICS on viral infection suggested that budesonide curtails the excessive inflammation induced by rhinovirus infections in patients with asthma [19]; and suppresses coronavirus (HCoV-229E) replication and cytokine production [20]. In vivo studies suggested a possible protective role of ICS in SARS-CoV-2 infection by reducing ACE-2 (angiotensin-converting enzyme 2) and TMPRSS2 (transmembrane protease serine 2) expression; being their co-expression needed for the virus to enter the cell [21]. Moreover, early administration of inhaled budesonide was studied in adults with mild COVID-19 symptoms; showing a lower need of urgent medical care and reduced recovery time [22]. In keeping with this, Yu et al. [23] showed (for 4700 adult patients at a higher risk of complications) that inhaled budesonide administered to SARS-CoV-2-positive participants improves recovery time. On the other hand, an observational study on 818,490 people with asthma does not support a major role for regular ICS use in protecting against COVID-19-related death in people with asthma [24]. In keeping with this, a rapid systematic review [25], published during the first wave of COVID-19, could not definitively establish a beneficial or detrimental effect of regular ICS therapy on the course of acute respiratory SARS-CoV2 infection. Consequently, even if corticosteroids are widely used to modulate the inflammatory response, there is no conclusive evidence on the potential benefits for people at risk of SARS-CoV-2 infections.

Finally, with respect to the possible role of allergic inflammation, Jackson et al. [26] showed that the expression of ACE2, needed for SARS-CoV-2 infection, was lower in patients with both high IgE levels and asthma. On the other hand, Sajuthi et al. [27] showed in 695 children that T2 inflammation reduced ACE2 expression and increased TMPRSS2 expression; thus having both a protective and a harmful role. Moreover, eosinopenia was considered as a biomarker of a poor prognosis [28]; strengthening the hypothesis that eosinophilia, typical of allergic patients, could be protective.

## 5. Children with CLD and Organization of Patient Care during COVID-19 Pandemic

After the spread of the COVID-19 pandemic, hospital and primary health care underwent major changes in clinical practice; mainly in order to reduce the risk of spreading the infection in health care providers and patients (Figure 1). During the first wave of the pandemic, most of the outpatient and inpatient services were temporarily suspended. An online survey [29] completed by ninety-one experts (caring for an estimated population of more than 133,000 children with asthma) showed that COVID-19 significantly reduced pediatric asthma services in the first months of 2020: 39% ceased in-person appointments, 47% stopped accepting new patients, and 75% limited patients’ visits. Moreover, Lubrano et al. [30], in a survey involving the Italian pediatric scientific societies, showed that during the first wave in 2020, many patients did not show up at their scheduled specialist visits for the fear of SARS-CoV-2 infection. In keeping with this, an observational study [31] at a tertiary care center of pediatric pulmonology in Ankara (Turkey) between January 2019 and December 2020, showed that the ongoing pandemic affected the routine clinical follow-up and pediatric pulmonology procedures; this was especially the case in non-COVID services. More specifically, in this report, outpatient visits decreased by 42.2% in 2020 compared to 2019; pulmonary function tests by 87.2%; and flexible bronchoscopy by 59.1% [31].

Lung function tests, indeed, were temporarily stopped in all the pulmonology services at the beginning of the pandemic because of the high risk of infection during expiratory maneuvers. Nonetheless, after a few weeks, the scientific societies for respiratory diseases [32] diffused recommendations for the safe performance of a lung function test; it could then start over.

Interestingly, during the pandemic, a rapid increase of telemedicine was registered all over the world; as it was considered an easy and effective tool to guarantee the continuity of care, avoiding access to healthcare facilities [33,34,35]. As an example, in March 2020, a pediatric infectious disease telemedicine program was developed and activated at an urban academic medical center in Parma (Italy); it was based on virtual visits with a real-time interaction, which enabled hospital access to be prevented in 90% of cases [33]. Recently, Hatziagorou et al. [36] described the effectiveness of telemedicine in their CF center, in which the onsite visits were replaced by a telephone visit call; this was considered a cheap, simple, and easily applicable solution with significant beneficial health effects in children with chronic diseases. Apart from the COVID-19 pandemic, telemedicine could have a role in the management of children with CLD; enabling a closer follow-up during which in-person and remote visits could alternate.

Nonetheless, further studies are needed to evaluate patient outcomes when moving from face-to-face visits to telemedicine; and to ask for patients’ opinion on this new modality. It is important to consider that telehealth could increase health care disparities, especially for those patients who may not have internet access. In a study analyzing the use of telemedicine in CF patients [37], authors described several barriers that could be overcome by expanding access to broadband internet services; promoting health and technology literacy; distributing devices that enable the remote monitoring of physiologic and psychologic parameters; and optimizing telehealth platforms to preserve the multidisciplinary approach.

## 6. Children with CLD and Psychosocial Impact of COVID-19 Pandemic

The SARS-CoV-2 pandemic, especially during the lockdown, affected children also from a psychological standpoint; this was mainly because of the burden of anxiety and depression [38]. Social distancing and all the protection measures imposed to avoid the spread of infection dramatically changed our lives; and threatened the mental health of children and their parents [39,40].

Children with CLD, because of the fear of having a poor outcome in the case of infection, were potentially more vulnerable to the psychological impact of pandemic-related restrictions [39,41,42]. Nonetheless, at the end of 2020, a prospective Italian study showed that PCD patients suffered from a psychological distress similar to controls; likely because of a low rate of pulmonary exacerbation during follow-up [43]. In keeping with this, a recent study assessed stress and mental health in adolescents, and young adults with Cystic Fibrosis; the study showed that the depression and anxiety scores were similar before and after the pandemic [44].

During the COVID-19 pandemic, changes in daily routine, with the closure of schools and the withdrawal of out-of-school activities, were responsible for the increase of loneliness among children and adolescents who ended up spending more time on digital devices [45]; and for the increase of sleep disorders [46].

A recent systematic review and meta-analysis [47], published up to September 2021, summarized the existing evidence from 30 studies on the link between mental health and digital media use in adolescents during the COVID-19 pandemic. The key message of the review is that, although the one-to-one or one-to-few communication based on social media is capable of reducing the feeling of loneliness, the general passive use of social media is clearly associated with diminished well-being [35]. Moreover, the increase of the hours spent in front of a television, smartphones, or tablets, dramatically reduced children’s physical activities with a detrimental effect on their general well-being; especially in children with CLD [48] who are used to, also in a non-pandemic period, having a slight reduction of physical activities in comparison with healthy controls [49].

## 7. Conclusions

Although COVID-19 rarely presents with a severe course in children, CLD may represent a risk factor; especially when already clinically severe or poorly controlled before SARS-CoV-2 infection. Therefore, children with CLD should be considered a priority for vaccination against SARS-CoV-2. On the other hand, the typical features of children with CLD (e.g., the accurate adoption of prevention measures) and notably, with asthma (e.g., the regular use of ICS, T2, and eosinophil-driven inflammation), might have a role in preventing SARS-CoV-2 infection.

As for health care services, during the pandemic, a rapid increase of telemedicine was registered all over the world. This approach has pros and cons and further studies are needed to evaluate whether it can have a role in the management of children with CLD; enabling a closer follow-up during which in-person and remote visits could alternate.

Finally, the SARS-CoV-2 pandemic significantly affected children and adolescents, including those with CLD, from a psychological standpoint. More particularly, the restrictions imposed to avoid the spread of infection brought a dramatic change in daily routine; limitations to social interaction with peers; interruption of in-person scholastic and recreational activities for variable periods; and an exponential increase in the use of digital devices.

## Figures and Tables

**Figure 1 ijerph-19-11483-f001:**
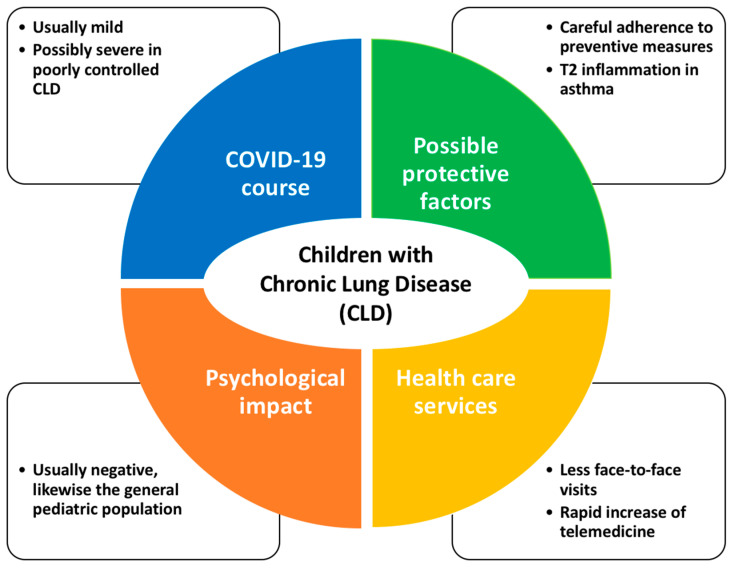
Features associated with COVID-19 in children with CLD.

## Data Availability

No new data were created in this study. Data sharing is not applicable to this article.

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
