# Peer review of "Impact of COVID-19 in Children with Chronic Lung Diseases"

_ijerph, 2022, doi:10.3390/ijerph191811483_

Round 1

Reviewer 1 Report

This is a well written review and the figure resumes very nicely the topic.

I have minor comments/suggestions.

In the section on psychosocial impact of Covid authors may consider to add the findings of the recent paper on mental health of CF adolescents (graziano s Ped Pulm 2022) further highlighting that people with chronic daises had minor impact from the pandemics (as for PCD).

In the conclusions adding a statement on the importance of vaccination against Covid for children with CLD might be worthy.

Author Response

R: We thank the Reviewer for his/her constructive comments. In the new version of the manuscript, we added the recent paper recommended by the author (ref 44; line 201-204 p. 5) and we added a statement regarding the importance of vaccinations against SARS-Cov-2 (line 106-107 p. 3; line 222-223 p.5).

Reviewer 2 Report

In this review, the authors examine the impact of COVID-19 on children with chronic lung diseases.

The title is short and apposite, the abstract is clear and appropriate in lengh and content and the keywords are well chosen.

The introdution is brief but sufficient to provide the necessary background.

The references are not soo many (considering this is a review) but they relevant and recent.

The choice of studies and the description of their results is good enough, so as the organization of the paper.

The conclusions are not so interesting, but anyway they are clearly described and well supported by the cited articles.

The scientific soundness and significance of content are average good, the originality is not so high but it's increased by the particular focus chosen, so this review may be of some interest for the readers of this journal.

Everything considered, in my opinion this review is suited for publication.

Author Response

R. We thank the Reviewer for his/her comments.